# The Non-Muscle-Splitting Mini-Incision Donor Nephrectomy Remains a Feasible Technique in the Laparoscopic Era of Living Kidney Donation

**Lex J. M. Habets** [1,2], **Andrzej G. Baranski** [1,2], **Khalil Ramdhani** [1], **Danny van der Helm** [2], **Ada Haasnoot** [1,2], **Aiko P. J. de Vries** [2,3], **Koen E. A. van der Bogt** [1,2], **Andries E. Braat** [1], **Jeroen Dubbeld** [1], **Hwai-Ding Lam** [1,2], **Jeroen Nieuwenhuizen** [1,2], **Willemijn N. Nijboer** [1,2], **Dorottya. K. de Vries** [1,2], **Ian P. J. Alwayn** [1,2], **Alexander F. M. Schaapherder** [1,2] and **Volkert A. L. Huurman** [1,2,*]

1   Department of Surgery, LUMC Transplant Center, Leiden University Medical Center, 2333 AZ Leiden, The Netherlands
2   LUMC Transplant Center, Leiden University Medical Center, 2333 AZ Leiden, The Netherlands
3   Department of Nephrology, LUMC Transplant Center, Leiden University Medical Center, 2333 AZ Leiden, The Netherlands
*   Correspondence: v.a.l.huurman@lumc.nl

**Abstract:** Laparoscopic donor nephrectomy (LDN) is the current gold standard in kidney donation. Mini-incision open donor nephrectomy (MINI) techniques have been used extensively but have become less popular. The aim of the present study was to compare the results and safety of a non-muscle-splitting MINI technique with the current gold standard of LDN. A single center retrospective cohort study of all living donor nephrectomies between 2011 and 2019 was used for the study. The primary outcome of this study was short term (<30 days) with Clavien–Dindo grade complications. Secondary outcomes included multivariable regression analysis of perioperative data. No differences in complication rates were observed between MINI and LDN and also after correction for known confounders. As expected, the operative time and first warm ischemia were significantly shorter in the MINI group and less blood loss was observed in the LDN group. Complications and conversion rate (LDN to open) among the LDN patients were in line with recent published meta-analyses. This study confirms the perioperative safety of living kidney donation in modern practice. Complication rates of both MINI and LDN procedures are limited and not different between procedures. In specific circumstances, the MINI procedure can still be considered a safe and feasible alternative for living kidney donation.

**Keywords:** living kidney donation; non-muscle-splitting mini-incision; laparoscopic donor nephrectomy

## 1. Introduction

In 2018, 51% of the transplanted kidneys in the Netherlands were retrieved from a living donor [1]. Kidney transplantation using living donor kidneys has been shown to be advantageous compared with postmortem-donor-derived kidneys for recipient outcome. Living donation provides the ability to better schedule the transplantation, limit graft ischemia times, select donors for optimal results, and requires less waiting time compared with postmortem donor waiting lists. Altogether, this leads to prolonged and increased graft survival after living donor kidney transplantation [2]. Living kidney donation, however, is a remarkable surgical procedure since it is performed in healthy individuals. Ensuring the safety of the procedure for the donor is therefore essential and should be the main concern when performing a living donor nephrectomy.

The first open donor nephrectomy was performed in 1954 by Murray et al. [3]. Since then, the open surgical approach has evolved from a large lumbotomy incision to slightly less invasive procedures with small incisions and less morbidity [4]. Since the introduction

of laparoscopic donor nephrectomy (LDN) in 1995 [5], and more recently robot-assisted nephrectomy [6], living donor nephrectomy has become increasingly popular. LDN has shown improved short-term results for the donors in comparison with the classic open approach [7–9]. In a Cochrane systematic review and meta-analysis published in 2011, it was shown that LDN was associated with less postoperative pain, reduced analgesia use, shorter length of hospital stay, and earlier return to normal function compared with open donor nephrectomy. [10] Therefore, either hand-assisted, transperitoneal, or retroperitoneal LDN has become the golden standard in living kidney donation. Most recent developments include a shift towards robot-assisted donor nephrectomy, although a clear benefit of this procedure still has to be proven.

At our center, a mini-incision donor nephrectomy technique (MINI) has been in use since 1990. This exact technique has not been described earlier and is unique in its vertically oriented, non-muscle splitting, extraperitoneal approach through the anterior abdominal wall. The abdominal muscles are not split, in contrast to other mini-incision approaches. The muscles are purposely left intact in order to minimize surgical trauma. Upon emerging evidence on the superiority of LDN, the mini-open technique was almost completely replaced by LDN. However, the specific MINI technique used at our center potentially holds advantages over other mini-open techniques, and has not been formally compared with LDN.

In some cases of kidney donation, LDN may not possible or desirable, for example, after earlier abdominal surgery or complex anatomy. Moreover, learning curves for MINI may be shorter than for LDN. Therefore, it is important to study the safety and feasibility of MINI in everyday surgical practice. Furthermore, the literature regarding superiority of LDN was mainly generated in the era prior to enhanced recovery after surgery (ERAS) programs that mainly benefit open procedures The positive effect of LDN on recovery in the current ERAS era may therefore be less evident.

The rationale for the current study was to determine the position of our well-developed non-muscle-splitting open approach for living donor nephrectomy when compared with the current LDN gold standard.

## 2. Materials and Methods

### 2.1. Patients

Between 2011 and 2019, 642 living kidney donors underwent either a MINI or LDN at the Leiden University Medical Center. This study period was chosen to minimize time-related and learning-curve-related effects. Within this period, no significant changes in the perioperative protocols were made and both procedures were performed by surgeons with ample experience in both surgical techniques.

In general, only patients with a Cockroft–Gault creatinine clearance > 80 mL/min were considered fit for donation. All donors were preoperatively screened by anesthesiologists and, if necessary, also by cardiologists. The responsible surgeon determined whether the left or right kidney was most suitable for donation based on the findings on CT scan with respect to the renovascular anatomy. In the case of a singular vascular anatomy, typically the left kidney was preferred for donation due to the longer renal vein, simplifying the transplantation procedure. The right kidney was preferred in the case of more complex left-sided anatomy. Given other anomalies (graft size, presence of benign cysts), generally, the 'best' kidney remained in the donor. In case of doubt whether both kidneys were equally functional, a radioisotope scan using $^{99m}$Tc-mercaptoacetyltriglycine (MAGIII) was performed, and when function was not equally distributed only the least active kidney was considered for donation. When low renal function or other general contraindications were expected, the donor was rejected for surgery. In some instances, primarily MINI was chosen because of complex vascular anatomy or surgical history. In other cases, both MINI and LDN procedures were discussed as options for surgery. The choice for either technique was dependent on both the patient's and the surgeon's preference (Supplementary Materials, Table S1).

Donors were admitted to the transplant ward one day prior to surgery. Standard bowel preparation was not performed. The donors received a 125 mL/h saline infusion in the 12 h before surgery to maintain sufficient hydration. All donors were allowed to eat until 6 h and drink until 2 h before surgery. A standard dose of 2850 IU of low molecular weight heparin was given the day and the morning before the surgery and continued until the day of discharge. Donors were discharged postoperatively when they were in good health and had resumed their normal diet. Follow-up visits were scheduled after 2 weeks with the nurse specialist on living donation, after 6 weeks at the nephrology and surgery outpatient clinic, and on a yearly basis afterwards to monitor kidney function.

The study was conducted in accordance with the Declaration of Helsinki and approved by the local research committee. All data were anonymized and stored according to local standards.

### 2.2. Non-Muscle-Splitting Mini Open Donor Nephrectomy (MINI)

Prophylactic antibiotics were administered to all donors prior to the surgical procedure. The antibiotic of choice was consistently cefazolin. The surgical technique of the MINI is described below.

With the patient placed in the mild Trendelenburg position, a skin incision is made along the lateral side of the rectus abdominis muscle starting about 2–3 cm below the costal arch for about 10 cm. Subcutaneous fat is cut until the anterior rectus sheath is reached. The anterior rectus sheath is opened/cut vertically starting 1 cm below the costal arch towards the linea semilunaris (Figure 1A). The lateral and posterior side of the rectus muscle is freed from small vessels and nerves and mobilized from the posterior sheath and gently pulled towards the medial side of the abdominal wall (Figure 1B). Note that the anterior rectus muscle is not split, in contrast to other open nephrectomy techniques. The main vascularization of the muscle is left intact. The lower part of this pararectal incision is retracted downwards to visualize the border between the peritoneum and linea semilunaris. From this point, the peritoneum is released from the abdominal wall and moved to the medial side in order to reach/visualize the retroperitoneal space (Figure 1C).

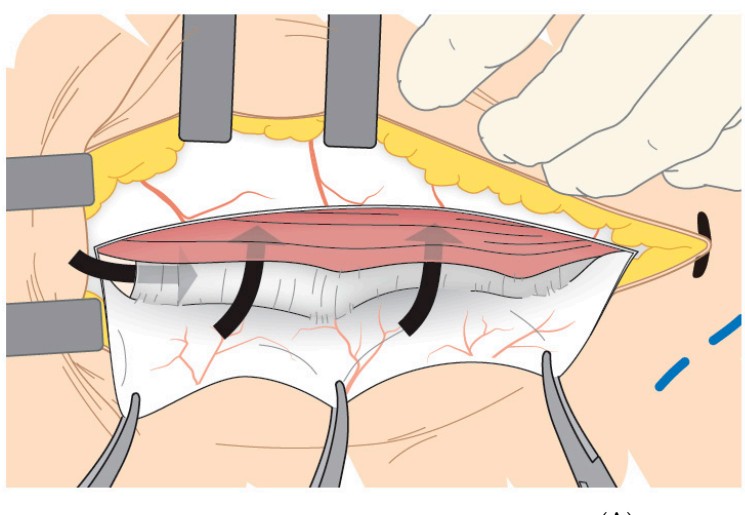 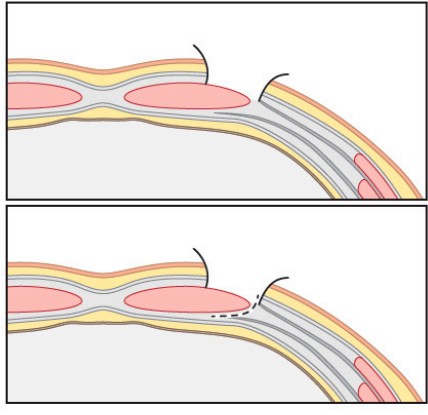

(**A**)

**Figure 1.** *Cont.*

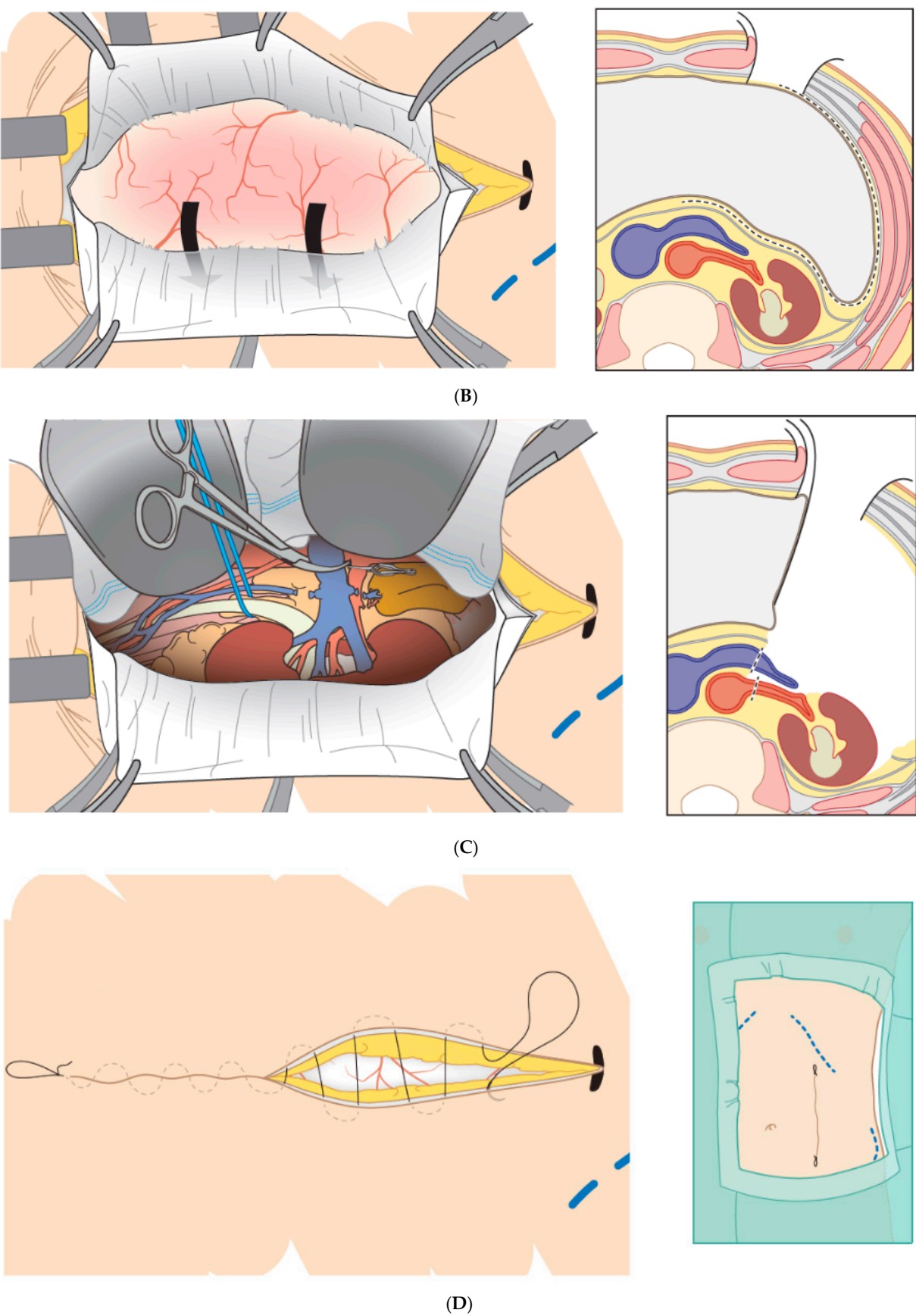

**Figure 1.** (**A**) A pararectal incision is made and the anterior rectus sheath is mobilized. (**B**) The peritoneum is dissociated from the abdominal wall to visualize the retroperitoneal space. (**C**) Gerota's fascia is opened and the kidney is dissected. (**D**) The skin is sutured intracutaneously.

After mobilization of the ureter and surrounding tissues, the gonadal vessels are ligated on both sides and cut above the iliac vessels. Gerota's fascia is then opened on the lateral side of kidney. After dissection of the kidney from the perinephric fat, the ureter, renal vein and renal artery are identified and encircled with vessel loops. The vessels are dissected and typically in left kidneys, the adrenal and gonadal veins are ligated and cut. After adequate renal vein and artery length is reached, the ureter is cut. The renal artery and vein are clipped, cut at adequate length for implantation (in general, the left vein is cut distal to the gonadal vein stump, and the right vein at its insertion in the caval vein), and oversewn using prolene suture. After retrieval, the kidney is perfused at the back table with cold preservation solution (Custodiol® HTK, Hong Kong, China) until all blood is washed out, and temporarily stored on ice. After evaluating hemostasis and insertion of a silicone drain, the posterior sheath, anterior sheath and subcutaneous fascia are closed with continuous running sutures. The skin is sutured intracutaneously (Figure 1D).

### 2.3. Laparoscopic Donor Nephrectomy (LDN)

LDN was performed according to international standards, as described below.

Similar pre-operative measures are taken as with the MINI procedure. In the operating theatre, the patient is placed in a 75-degree lateral bent position. A subcostal midclavicular 12 mm trocar and port are introduced. After CO2 insufflation of the abdomen, 3 or 4 additional trocars are introduced. Subsequently, the hepatic or splenic flexure of the colon are mobilized to the medial side following the caudal identification of the m. psoas and ureter. In left kidney donation, the gonadal and adrenal veins are identified and ligated/clipped. Gerota's fascia is now opened and the renal vasculature is identified and dissected. A Pfannenstiehl suprapubic incision is made and the renal artery and vein are transected using endostaplers (TA30 endovascular, Covidien, Dublin, Ireland). In most cases, the kidney is placed inside an endobag (Inzii, Applied Medical, Rancho Santa Margarita, CA, USA) and extracted through the suprapubic incision. Depending on the surgical situation, a hand may be placed through the incision purely for kidney retrieval. After removal, the organ is perfused and stored as described for MINI. Before closing the abdominal wall at the extraction site, hemostasis is checked and the colon is placed in its normal position. After a final hemostasis check, the pneumoperitoneum is deflated and the fascia is closed. The skin is sutured intracutaneously. In a small minority of LDN, kidneys are retrieved using a hand-assisted retroperitoneoscopic (HARP) technique, solely based on surgeon preference [8].

### 3. Outcome Measures

To evaluate the short-term outcome of the MINI technique, the possible association with several outcome parameters was analyzed and compared with the current gold standard of LDN. Primary outcome for this analysis was short-term (<30 days) complication rate, as graded by the Clavien–Dindo score (CDS) [11]. Clavien–Dindo score complications grade 1 and 2 were considered minor, whereas 3a and higher were considered major. Secondary outcome measures included operation time (OT), warm ischemic time (WIT), estimated blood loss (EBL), 1-year donor kidney function, and 1-year complication rate.

### 4. Statistical Analysis

IBM SPSS Statistics version 25 was used for statistical analysis. For analysis of independent samples, Student's *t*-tests were performed. Categorical variables were compared using Chi-squared or Fischer's exact tests. Continuous data are presented as mean with standard deviation. Multivariable linear regression with a forward selection of variables was used to assess variables independently associated with complication grade, OT, EBL, and first WIT. The significance level to enter the multivariable model was set to $p = 0.20$ and $p$-values $< 0.05$ were considered statistically significant.

## 5. Results

### 5.1. Patient Preference and Previously Undergone Surgical Treatments

A total of 233 (36.3%) patients had a history of abdominal surgery. No differences were found in the number and type of previous surgeries between the MINI and LDN group (104 vs. 129, *p* = 0.666). Patient preference regarding MINI or LDN was documented in 402 (62.4%) of the cases (Supplementary Materials, Table S1).

### 5.2. Donor Characteristics

A total of 642 living kidney donors were included in the present study (287 MINI, 355 LDN). Mean age of the donors was 53 years (SD ± 12) with a mean BMI of 25.9 kg/m$^2$ (SD ± 3.6). In our cohort, 83 (12.9%) donors were obese (BMI > 30 kg/m$^2$) while 8 had a BMI greater than 35 kg/m$^2$. In total, 59% of donors were female. No significant differences were found in BMI between both procedure groups (MINI: 26.0, LDN: 25.7). Pre-donation eGFR was also comparable (MINI: 57.1 umol/L, LDN: 56.6 umol/L), as was the number of left-sided nephrectomies (MINI: 83.6%, LDN: 83.1%) (Table 1). Hand assisted donor laparoscopy was performed in 117 (30.1%) of the cases.

**Table 1.** Non-Muscle-Splitting MINI and LDN donors.

| | MINI (N = 287) | LDN (N = 355) | *p*-Value |
|---|---|---|---|
| Donor Related | | | |
| Age (years) | 55 ± 11 | 52 ± 12 | 0.002 |
| Sex (male/female) | 50.2/49.8% | 33.8/66.2% | <0.001 |
| Length (cm) | 173 ± 9.5 | 172 ± 9.5 | 0.068 |
| Weight (kg) | 78.7 ± 13.2 | 76.7 ± 13.7 | 0.050 |
| BMI | 26.0 ± 3.6 | 25.7 ± 3.7 | 0.257 |
| Pre-donation Creatinine (μmol/L) | 73.8 ± 12.5 | 74.0 ± 40.3 | 0.936 |
| Operation Related | | | |
| WIT (s) | 100 ± 56 | 296 ± 117 | <0.001 |
| Operation duration (min) | 175 ± 40 | 187 ± 41 | <0.001 |
| Estimated blood loss (mL) | 275 ± 382 | 136 ± 201 | <0.001 |
| Hand assisted | - | 107 (30.1%) | - |
| Converted to hand assisted | - | 6 (1.7%) | - |
| Converted to open | - | 3 (0.8%) | - |
| Donation period | | | <0.001 |
| 2011–2013 | 113 (58.3%) | 81 (41.7%) | |
| 2014–2016 | 132 (61.4%) | 83 (38.6%) | |
| 2017–2019 | 22 (10.3%) | 191 (89.7%) | |
| Hospitalization (days) * | 3.2 ± 1.6 | 3.1 ± 1.1 | 0.504 |
| Follow up length (years) | 3.6 ± 2.4 | 2.3 ± 2.4 | <0.001 |
| Graft Related | | | |
| Number of Arteries (%) | | | 0.114 |
| 1 | 232 (81%) | 300 (85%) | |
| ≥2 | 55 (19%) | 54 (15%) | |
| Number of Veins (%) | | | 0.308 |
| 1 | 276 (96%) | 345 (97%) | |
| ≥2 | 11 (4%) | 10 (3%) | |
| Right kidneys (%) | 46 (16.4%) | 60 (16.9%) | 0.426 |
| Pre-donation eGFR (mL/min/1.73 m$^2$) | 89.4 ± 17.2 | 90.3 ± 17.1 | 0.492 |
| 1 year after donation eGFR (mL/min/1.73 m$^2$) | 57.1 ± 12.0 | 56.6 ± 14.8 | 0.648 |

\* Days between donation date and discharge from hospital. Data are represented as mean and standard deviation or as percentages.

Statistically significant differences between the two study groups were found in gender and age of the donors. In the MINI group, mean age was 55 years (SD ± 11) compared with 52 years (SD ± 12) in the LDN group (*p* = 0.002). A smaller proportion of women underwent MINI compared with LDN (50% vs. 66%, *p* ≤ 0.001). The yearly number of living donor nephrectomies largely remained unchanged over the study period. However,

the surgical technique of choice did change significantly over time ($p \leq 0.001$). MINI was performed in 58.3% of the procedures between 2011–2013, which decreased to 10.3% between 2017–2019 (Table 1).

### 5.3. Primary Outcome

No donors died in the perioperative period. A total of 71 perioperative complications were observed in 67 donor procedures (11.1%). Four donors experienced multiple complications. The type and severity of perioperative complications were graded according to the CDS classification (Table 2). The most common types of complications were bowel or spleen lacerations (14% of 71 complications, one of which was graded as major in the MINI group), hospital-acquired pneumonias (14%, all considered minor), and wound infections (14%, all considered minor). In nine LDN donors (2.5%), the surgical team converted to a hand-assisted nephrectomy. Iatrogenic bowel injury was most common in the LDN group, whereas splenic lacerations were limited to the MINI group. In one such MINI case, splenectomy had to be performed due to the severity of the injury.

**Table 2.** Short-term complications incidence by type, graded by the Clavien–Dindo score.

| | MINI (N = 287) | LDN (N = 355) |
|---|---|---|
| **No Complications** | **262 (91.3%)** | **314 (88.5%)** |
| **Grade 1/2 (Minor)** | | |
| Hematoma | 0 | 3 |
| Wound infection | 5 | 5 |
| Obstipation | 0 | 1 |
| Perioperative bleeding | 0 | 2 |
| Postoperative bleeding | 1 | 0 |
| Iatrogenic bowel injury *w/o* consequence | 0 | 5 |
| Iatrogenic ureter injury *w/o* consequence | 0 | 1 |
| Iatrogenic splenic injury | 3 | 0 |
| Iatrogenic kidney injury | 0 | 1 |
| Gastritis | 0 | 1 |
| Testicular pain | 0 | 3 |
| Bleeding out of meatus | 0 | 1 |
| Kidney function disorder | 0 | 1 |
| Pancreatitis | 0 | 1 |
| Postoperative neuropraxia | 1 | 1 |
| Chylus leakage | 0 | 1 |
| Retention of urine | 1 | 0 |
| Acute kidney disorder | 2 | 0 |
| Perioperative bleeding: transfusion | 1 | 1 |
| Pneumonia | 6 | 4 |
| Urinary tract infection | 1 | 2 |
| Pulmonary embolism | 0 | 2 |
| Hypertension | 0 | 3 |
| Deep venous thrombosis | 0 | 1 |
| Epididymitis | 0 | 1 |
| Wound dehiscence | 1 | 0 |
| *Subtotal* | *22 (7.7%)* | *41 (11.6%)* |
| **Grade 3a/b (Major)** | | |
| Pancreatic leakage | 1 | 0 |
| Splenectomy due to injury | 1 | 0 |
| Genitofemoral reintervention | 0 | 1 |
| Severe rebleeding | 1 | 0 |
| Incisional hernia | 0 | 1 |
| Re-exploration of scar | 0 | 1 |
| Chylus leakage | 0 | 1 |
| *Subtotal* | *3 (1.0%)* | *4 (1.1%)* |

Multiple complications occurring in a single patient were counted separately.

Most complications were considered minor (88.7%). A total of 7 donors (1.1% of 642, 3 in MINI group, and 4 in LDN group) experienced a major complication. All of these major complications required surgical reintervention within 30 days after donation. One MINI donor experienced a major complication in combination with another minor complication. Whereas three LDN donors had multiple minor complications, no multiple minor complications were found in the MINI group. The mean hospital stay among donors who experienced minor complications was 4 days (SD $\pm$ 2) while donors with major complications had a significantly longer hospital stay of 7 days (SD $\pm$ 7). Donors without complications stayed 3 days in the hospital. Of note, no further complications were reported in both groups from 30 days to 1 year after surgery.

*5.4. Secondary Outcomes*

Mean operation time in the MINI group was 175 (SD $\pm$ 40) minutes versus 187 (SD $\pm$ 40) minutes ($p \leq 0.001$) in the LDN group (Table 1). Mean first warm ischemia time was 100 (SD $\pm$ 56) seconds in the MINI group compared with 296 (SD $\pm$ 117) seconds in the LDN group ($p \leq 0.001$). Estimated blood loss was 275 (SD $\pm$ 382) mL in MINI compared with 136 (SD $\pm$ 201) mL in LDN ($p \leq 0.001$). No statistically significant difference was found in the number of hospitalized days (MINI: 3.2, LDN: 3.1, $p = 0.504$). In the LDN group, in 30.1% of the procedures the kidney was retrieved by hand. Furthermore, 2.5% of LDN procedures were converted to either hand-assisted or open procedures. Multiple arterial anatomy was found in 19% of MINI donors compared with 15% in LDN ($p = 0.114$). Estimated kidney function (eGFR) at 1-year post-transplantation was 57.1 (SD $\pm$ 12.0) in MINI versus 56.6 (SD $\pm$ 12.1) mL/min/1.73 m$^2$ in recipients of LDN donors ($p = 0.648$).

*5.5. Impact of MINI on Perioperative Parameters: A Multivariable Model*

To determine variables contributing to important perioperative outcomes such as operation time, blood loss, hospital stay, and complication grades, a multivariable regression analysis was performed (Table 3). Operation times were significantly longer in male donors ($p \leq 0.001$), donors having higher BMI ($p \leq 0.001$), donors who had kidneys with multiple arteries ($p \leq 0.001$), and donors undergoing nephrectomy using the LDN technique ($p \leq 0.001$). All these variables were both univariably and multivariably associated with longer operation time. Higher amounts of blood loss were also independently associated with the MINI technique ($p \leq 0.001$) and donor male gender ($p = 0.007$). The MINI technique was significantly associated with shorter first warm ischemic times ($p \leq 0.001$), whereas right sided kidneys ($p = 0.029$) and multiple arterial anatomy ($p \leq 0.001$) were associated with longer first warm ischemic times. In this model, none of the defined variables had significant associations with length of hospital stay and complication grades.

**Table 3.** Uni- and multivariable analysis.

| | Parameter | Univariable Analysis B (95% CI) | *p*-Value | Multivariable Model B (95%CI) | *p*-Value |
|---|---|---|---|---|---|
| Complication grade | Donor age | $-0.02$ ($-0.54$, 0.02) | 0.264 | - - - - | n.s. |
| | Donor male gender | $-0.02$ ($-0.93$, 0.89) | 0.97 | - - - - | n.s. |
| | Donor BMI | $-0.06$ ($-0.19$, 0.07) | 0.370 | - - - - | n.s. |
| | Multiple renal arteries | 0.34 ($-0.60$, 1.28) | 0.476 | - - - - | n.s. |
| | Right kidney | $-0.12$ ($-1.38$, 1.15) | 0.855 | - - - - | n.s. |
| | MINI technique | 0.07 ($-0.83$, 0.96) | 0.886 | - - - - | n.s. |
| Operation time | Donor age | $-0.31$ ($-0.67$, 0.04) | 0.083 | 0.09 ($-0.17$, 0.35) | 0.480 |
| | Donor male gender | 21.30 (12.99, 29.60) | <0.001 | 23.1 (16.9, 29.2) | <0.001 |
| | Donor BMI | 1.85 (0.97, 2.72) | <0.001 | 1.90 (1.07, 2.73) | <0.001 |
| | Multiple renal arteries | 14.36 (6.95, 21.77) | <0.001 | 12.90 (5.21, 19.39) | 0.001 |
| | Right kidney | $-4.72$ ($-13.34$, 3.90) | 0.283 | - - - - | n.s. |
| | MINI technique | $-12.25$ ($-18.62$, $-5.87$) | <0.001 | $-17.57$ ($-23.69$, $-11.44$) | <0.001 |

**Table 3.** *Cont.*

| | Parameter | Univariable Analysis B (95% CI) | *p*-Value | Multivariable Model B (95%CI) | *p*-Value |
|---|---|---|---|---|---|
| First warm ischemia time | Donor age | −1.60 (−2.57, −0.63) | 0.001 | −0.67 (−1.37, 0.04) | 0.063 |
| | Donor male gender | 6.07 (−17.49, 29.64) | 0.613 | - - - - | n.s. |
| | Donor BMI | 3.32 (−3.47, 10.12) | 0.337 | - - - - | n.s. |
| | Multiple renal arteries | 19.43 (−7.55, 46.41) | 0.158 | 35.84 (16.18, 55.49) | <0.001 |
| | Right kidney | 29.29 (−1.62, 60.20) | 0.063 | 24.70 (2.53, 46.87) | 0.029 |
| | MINI technique | −195.1 (−212.5, −177.8) | <0.001 | −194.5 (−211.7, −177.3) | <0.001 |
| Estimated blood loss | Donor age | 0.18 (−1.90, 2.26) | 0.864 | - - - - | n.s. |
| | Donor male gender | 93.94 (44.91, 142.97) | <0.001 | 67.43 (18.78, 116.09) | 0.007 |
| | Donor BMI | 3.32 (−3.47, 10.12) | 0.337 | - - - - | n.s. |
| | Multiple renal arteries | 11.58 (−45.63, 68.78) | 0.691 | - - - - | n.s. |
| | Right kidney | 3.45 (−61.88, 68,77) | 0.918 | - - - - | n.s. |
| | MINI technique | 140.1 (92.4, 187.8) | <0.001 | 127.9 (79.7, 176.2) | <0.001 |
| Length of hospital stay | Donor age | 0.01 (−0.004, 0.014) | 0.245 | - - - - | n.s. |
| | Donor male gender | −0.08 (−0.30, 0.13) | 0.435 | - - - - | n.s. |
| | Donor BMI | −3.42 (−22.95, 16.10) | 0.731 | - - - - | n.s. |
| | Multiple renal arteries | −14.29 (−178.4, 149.8) | 0.864 | - - - - | n.s. |
| | Right kidney | 44.40 (−144.7, 233,5) | 0.645 | - - - - | n.s. |
| | MINI technique | 0.07 (−0.14, 0.28) | 0.487 | - - - - | n.s. |

## 6. Discussion

Living kidney donation is a known valuable addition to postmortem-donated kidneys, and has been able to increase the number of kidney transplants and reduce the time on the waiting list. In some countries, living donor kidneys comprise half of all transplanted kidneys, leading to excellent and longer graft and patient survival [10]. Although LDN has become the standard technique of choice, the MINI-open donor nephrectomy is still performed in many centers, especially in the case of complex vascular or abdominal anatomy. As MINI concerns an open surgical technique, learning curves may be shorter for centers setting up a living donor program. The goal of the present study was to objectively compare the perioperative and short-term outcomes between LDN and MINI techniques in the same period in the absence of a learning curve effect.

In our study, no difference was found between MINI and LDN complication rates. Although some baseline characteristics were different between the groups, these were not deemed of significant influence on outcome. As expected, operating time and first warm ischemia were significantly shorter in the MINI group whereas blood loss was less in the LDN group. The complication incidence and conversion rate among LDN patients were in line with multiple other recent studies [10,12–15]. In contrast to other literature [14–17], no prolonged hospitalization was observed in the MINI group compared with laparoscopic donors (3.2 days vs. 3.1 days, *p* = 0.504), indicating a comparable initial recovery period as has been shown in another recent study. [18] This finding may be related to the adoption of enhanced recovery after surgery programs for both open and laparoscopic surgery in the last decade [19].

This study has its limitations, as it is a single-center retrospective analysis without randomization of the surgical technique used. Some important factors such as post-operative pain were not studied [20]. However, post-operative pain was not the rationale nor an outcome parameter of the current retrospective analysis, which focuses on the feasibility of a known surgical technique. In the study period, both techniques were considered (and presented to patients) as of equal quality. Some patients had a specific desire for a certain technique, which was accommodated. If no specific desire was present, the attending surgeon chose the most familiar technique. This is also represented by the shift over time from MINI towards LDN, due to increased experience with LDN of mainly younger surgeons

joining the team. Although not randomized, the cohort therefore represents a 'real-life' clinical practice rather than a restricted study environment. In contrast to earlier, randomized, studies, complications and length of stay were not different in real life with similar modern peri-operative protocols and surgeons with ample experience in the technique used. This equality remained intact after correction for known confounders.

Considering the limitations of this study, it may be concluded that MINI is a well-developed and safe technique that in daily practice equals LDN in terms of complication rate and hospital stay.

As in most centers, at our center LDN has become the standard of care because of surgical developments, patient expectations, and evidence from the literature. However, we conclude from this analysis that MINI can still be safely used in environments or circumstances where the use of LDN is less desirable. This may be the case in centers with a lack of experience or resources, or in the case of non-standard surgical anatomy.

**Supplementary Materials:** The following supporting information can be downloaded at: https://www.mdpi.com/article/10.3390/transplantology4010001/s1, Table S1: Previous surgical treatments & patient and surgeon preferences.

**Author Contributions:** L.J.M.H. participated in research design, data collection, analysis, and writing the paper. A.G.B. participated in research design, data collection, analysis, and writing the paper. K.R. participated in data collection and research design. D.v.d.H. participated in data collection and analysis. A.H. participated in data collection. A.P.J.d.V. participated in data collection and analysis. K.E.A.v.d.B. participated in data collection and analysis. A.E.B. participated in data collection and analysis. J.D. participated in data collection and analysis. H.-D.L. participated in data collection and analysis. J.N. participated in data collection and analysis. W.N.N. participated in data collection and analysis. D.K.d.V. participated in data collection and analysis. I.P.J.A. participated in data collection and analysis and writing the paper. A.F.M.S. participated in research design, data collection, analysis, and writing the paper. V.A.L.H. participated in research design, data collection, analysis, and writing the paper. All authors have read and agreed to the published version of the manuscript.

**Funding:** This study received no external funding.

**Institutional Review Board Statement:** The study was conducted in accordance with the Declaration of Helsinki, and its protocol approved by the Institutional Review Board (or Ethics Committee) of the Leiden University Medical Center on 10 December 2019.

**Informed Consent Statement:** Informed consent from all patients for use of their perioperative data for quality assessment was obtained prior to surgery.

**Data Availability Statement:** Data regarding the current study can be requested from the LUMC Transplant Center Database.

**Acknowledgments:** The authors would like to thank Manon Zuurmond for providing the illustrations of the non-muscle-splitting MINI procedure.

**Conflicts of Interest:** The authors declare no conflict of interest.

## Abbreviations

| | |
|---|---|
| BMI | Body mass index |
| CDS | Clavien–Dindo score |
| CIT | Cold ischemic time |
| EBL | Estimated blood loss |
| EGFR | Estimated glomerular filtration rate |
| HAP | Hospital-acquired pneumonia |
| HARP | Hand-assisted retro peritoneoscopy |
| LDN | Laparoscopic donor nephrectomy |
| MINI | Non-muscle-splitting mini donor nephrectomy |
| OT | Operation time |
| WIT | Warm ischemic time |

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
