# Peer review of "The Non-Muscle-Splitting Mini-Incision Donor Nephrectomy Remains a Feasible Technique in the Laparoscopic Era of Living Kidney Donation"

_2673-3943, doi:10.3390/transplantology4010001_

Round 1
Reviewer 1 Report
Dear authors,
Interesting work showing that MINI probably is as good as LDN although groups were not randomized.
I have one issue that could add value to the paper. Ofcourse it is a pitty that painscores were not regularly reported or investigated. Length of hospital stay (equal in both groups) is only a surrogate measure for pain equivalence.
However patients were invited 2 and 6 weeks postoperatively at the outpatient ward. Were there any differences in the use of pain medication at 2 or 6 weeks? And maybe more important were there any differences in "return to work" or return to "daily activities" at both time intervals? I hope this can be found back in the files and added to the secondary outcomes as it may be important to know if both techniques are equal with respect to these midterm outcomes that are important to patients.
Another suggestion to improve overall readability is to merge Grade 1 and 2 complications (both are minor) and merge 3a and 3b complications (both major). This can be explained in the Methods section. Because now I see woundinfection and kidney disorder both in Grade 1 and grade 2 which is surrogate accuracy and difficult to interprete as for example no criteria are given to grade woundinfections.
Why is bowel or ureter injury minor? Was the bowel opened?
Minor comments:
- MINI does require splitting of the transverse rectus muscle, doesn't it?
- Why leave behind a drain in MINI? I think surgeons leave behind a perfectly dry operating field in such important interventions, so what is the added value of a drain?
- I would start the result section with this paragraph : (line 286-289)
Patient preference and previously undergone surgical treatments
A total of 233 (36.3%) patients had a history of abdominal surgery. No differences were found in the number and type of previous surgery between the MINI and LDN group (104 vs. 129, p = 0.666). Patient preference regarding MINI or LDN was documented in 402 (62.4%) of the cases (Supplementary Table 1).
Author Response
Thank you for these comments. Please find our answers below.
Regarding pain medication during postoperative outpatient visit:
In hindsight, it would have been good to determine pain scores on a routine basis both during hospital stay and during the outpatient visit. Unfortunately this was not the case. The vast majority was taking paracetamol or no pain medication at all, at the moment of outpatient visit. Chronic pain was observed very rarely. In general, regular work activities had already been resumed at the surgeon visit at 6 weeks. There is no possibility to retrieve these data at this point and it would be doubtful if any significant differences would be detected.
At the same time, this was not the main goal of this retrospective analysis. Its goal was to determine the place of MINI nephrectomy in the current laparoscopic landscape. Our center performs laparoscopic donor nephrectomy as a routine, but this analysis strengthens us that the MINI technique may be a good alternative in selected cases.
Regarding the complications table: this has been merged as suggested by the reviewer.
Bowel or ureter injury that was detected during the procedure and either small and repaired (bowel) or of no consequence for the procedure (ureter). This was graded as minor due to the lack of consequence, as determined in the Clavien-Dindo classification.
MINI does not require splitting of the transverse rectus muscle, as only its lateral conjoint tendon is split (as can be seen in Figure 1B). No muscles are split during this procedure.
Drains were routinely placed as part of the MINI surgeons’ routine. We agree with the reviewer that no clear evidence exists on its added value.
We have changed the start of the results section according to the reviewer’s suggestion (p 11 l 4-7).
Reviewer 2 Report
Thanks for giving me the opportunity to review this manuscript. Researchers need to justify the rationale to do this study when so much literature is available on this topic.
- DOI: 10.1016/j.ijsu.2018.04.003
https://clinicaltrials.gov/ct2/show/NCT00258986
https://www.ncbi.nlm.nih.gov/pmc/articles/PMC4089213/
Its mentioned that some of these studies were not done in ERAS era but designing a retrospective study where the pain and other ERAS outcomes cannot be evaluated will limit the logic to perform the study.
A strong introduction, rationale, and discussion are recommended.
Author Response
Thank you for these comments. The studies mentioned by the reviewer (one of them referenced) show similar results to our study, with the difference that our method does not involve muscle splitting at all, since a lateral route in between muscle groups is chosen (see Figure 1).
In our center, laparoscopic donor nephrectomy (LDN) is currently the method of choice, based on literature and patient and surgeon preference. The rationale of the current analysis was to determine the position of our locally well-developed mini-open technique in the current era. Taking into account the limitations of this analysis, we feel safe to conclude that this technique is feasible and safe, does not seem to be inferior to LDN in our hands, and can be considered as an alternative in selected cases.
To clarify this rationale further, we have included a specific rationale to the introduction (p 7 l 1-4).